# Upper Gastrointestinal Lesions during Endoscopy Surveillance in Patients with Lynch Syndrome: A Multicentre Cohort Study

**DOI:** 10.3390/cancers13071657

**Published:** 2021-04-01

**Authors:** Romain Chautard, David Malka, Elia Samaha, David Tougeron, Didier Barbereau, Olivier Caron, Gabriel Rahmi, Thierry Barrioz, Christophe Cellier, Sandrine Feau, Thierry Lecomte

**Affiliations:** 1Department of Hepatogastroenterology and Digestive Oncology, Trousseau University Hospital, CHU de Tours, CEDEX 09, 37044 Tours, France; d.barbereaupro@orange.fr (D.B.); sandrine.feau@hotmail.fr (S.F.); 2Gastrointestinal Oncology Unit, Department of Oncologic Medicine, Institut Gustave Roussy, Université Paris Sud, 94805 Villejuif, France; david.malka@gustaveroussy.fr (D.M.); olivier.caron@igr.fr (O.C.); 3Department of Gastroenterology, European Georges Pompidou Hospital, Assistance Publique Hôpitaux de Paris, 75015 Paris, France; elia.samaha@aphp.fr (E.S.); gabriel.rahmi@egp.aphp.fr (G.R.); christophe.cellier@egp.aphp.fr (C.C.); 4Department of Gastroenterology, Poitiers University Hospital, 86021 Poitiers, France; david.tougeron@chu-poitiers.fr (D.T.); t.barrioz@chu-poitiers.fr (T.B.)

**Keywords:** Lynch syndrome, upper gastrointestinal endoscopy, gastric cancer, duodenal cancer, screening, *Helicobacter pylori*

## Abstract

**Simple Summary:**

Patients with Lynch syndrome are at increased risk of upper gastrointestinal cancer. Recommendations for upper gastrointestinal endoscopy screening vary widely with limited data supporting effectiveness. The aim of our study was to investigate yields of upper gastrointestinal endoscopy screening in a large multicentre cohort of 172 Lynch syndrome mutation carriers. In our study, upper gastrointestinal endoscopy surveillance detects frequent neoplastic lesions particularly after the age of 40 years. Ours results suggest that Lynch patients should be considered for upper gastrointestinal endoscopic and *Helicobacter pylori* screening.

**Abstract:**

Background: Patients with Lynch syndrome are at increased risk of gastric and duodenal cancer. Upper gastrointestinal endoscopy surveillance is generally proposed, even though little data are available on upper gastrointestinal endoscopy in these patients. The aim of this retrospective study was to evaluate the prevalence and incidence of gastrointestinal lesions following upper gastrointestinal endoscopy examination in Lynch patients. Methods: A large, multicentre cohort of 172 patients with a proven germline mutation in one of the mismatch repair genes and at least one documented upper gastrointestinal endoscopy screening was assessed. Detailed information was collected on upper gastrointestinal endoscopy findings and the outcome of endoscopic follow-up. Results: Seventy neoplastic gastrointestinal lesions were diagnosed in 45 patients (26%) out of the 172 patients included. The median age at diagnosis of upper gastrointestinal lesions was 54 years. The prevalence of cancer at initial upper gastrointestinal endoscopy was 5% and the prevalence of precancerous lesions was 12%. Upper gastrointestinal lesions were more frequent after 40 years of age (*p* < 0.001). *Helicobacter pylori* infection was associated with an increased prevalence of gastric, but not duodenal, lesions (*p* < 0.001). Conclusions: Neoplastic upper gastrointestinal lesions are frequent in patients with Lynch syndrome, especially in those over 40 years of age. The results of our study suggest that Lynch patients should be considered for upper gastrointestinal endoscopic and *Helicobacter pylori* screening.

## 1. Introduction

Lynch syndrome (also known as hereditary nonpolyposis colorectal cancer (HNPCC) syndrome) is an autosomal dominant syndrome involving germline mutations in genes that encode DNA mismatch repair (MMR) proteins [1]. Inactivation of the DNA MMR genes (*MLH1, MSH2*, or more rarely *MSH6* and *PMS2*) predisposes carriers to multiple malignancies, including early-onset colorectal and endometrial cancer, or less frequently ovarian, small bowel, urothelium, biliary tract, and gastric cancers [1,2]. Lynch syndrome is responsible for approximately 2% of all diagnosed cases of colorectal cancer [3]. Individuals with Lynch syndrome have about a 50–80% lifetime risk of colon cancer, and women with Lynch syndrome have a 60% lifetime risk of endometrial cancer [4].

Gastric cancer has been reported to be the third or fourth most common extracolonic cancer in Lynch syndrome. The cumulative lifetime risk of gastric cancer in Lynch syndrome (LS) is reported to be varying from 2% to 8% [2,5,6,7]. This risk is highest in older patients and those with the pathogenic MSH2 and MLH1 variants [7]. Even though the relative risk of developing small bowel carcinoma in Lynch syndrome is extremely high (over 100), the lifetime risk remains relatively low (2–8%), consisting of duodenal carcinomas in approximately half of the cases [6,8,9,10,11]. However, a recent large study reported that the estimated cumulative risk for gastric cancer and small bowel cancer is less than 1% at 70 years, which is lower than previously reported [8].

On the basis of observational studies, colonoscopy screening has been shown to be effective in reducing the incidence and mortality of colorectal cancer among MMR mutation carriers [12]. Colonoscopy screening is recommended every 1 to 2 years, beginning at age 20–25, or 5 years younger than the earliest-onset cancer recorded among the affected members in the family [13,14]. Although few surveillance data for gastric or small-bowel cancer are available, screening individuals with Lynch syndrome using upper gastrointestinal endoscopy (UGE) is recommended by several expert consensus statements, but it remains controversial [13,14,15,16,17,18]. *Helicobacter pylori* (*H. pylori*) infection is associated with increased risk of gastric cancer [19,20]. *H. pylori* eradication prevents gastric cancer and screening is recommended in patients at high risk of gastric cancer [21]. The sequence of events from *H. pylori* infection, through atrophic chronic gastritis with metaplasia to dysplasia, is considered to be the precursor cascade of gastric cancer, especially of an intestinal type [20]. In Lynch syndrome, the intestinal type seems the most frequent form of gastric cancer [5,22].

The goal of UGE is to detect precancerous lesions at an early curable stage and to detect *H. pylori* infection. Only few studies have evaluated the prevalence of *H. pylori* infection and the effectiveness of surveillance for gastric cancer in individuals with Lynch syndrome [23,24,25,26,27]. Therefore, the utility and optimal modalities for screening for gastric and duodenal cancers in individuals with Lynch syndrome remain undefined. In spite of these uncertainties, guidelines from numerous professional societies recommend consideration of UEG screening with esophagogastroduodenoscopy and testing for *H. pylori* for all individuals with LS (Table 1).

Recent guidelines recommend that UGE surveillance in Lynch syndrome should only be performed in the context of a clinical trial [14]. However, there is a general consensus on the need to screen *H. pylori* in patients with Lynch syndrome and subsequent eradication therapy. We report here the incidence and prevalence of gastrointestinal lesions and *H. pylori* infection in a large, retrospective, multicentre cohort of individuals with Lynch syndrome.

## 2. Patients and Methods

### 2.1. Study Design

Individuals were enrolled at four tertiary-care centres experienced in Lynch syndrome management. Individuals were eligible for this study if they had a proven pathological germline mutation in one of the *MMR* genes (*MSH2*, *MLH1*, *MSH6,* or *PMS2*) and with at least one UGE screening. The primary endpoint was the proportion of abnormal UGE among the screened patients. At our centres, we routinely perform UGE surveillance in Lynch syndrome. Experienced endoscopists performed UEG screening with standard adult endoscopes. Systematic biopsy was not performed during the UEG screening and was left to the discretion of the endoscopist. This study was reviewed and approved by the local Ethics Committee (Comité de Protection des Personnes, Tours, France, Number 20021 004).

### 2.2. Subjects and Data Collection

Patients were included and data were recorded by each centre retrospectively and anonymously between December 2009 and May 2011. The data collected included gender, date of birth, previous history of cancer, family history of gastric and duodenal cancers, type of MMR gene mutation, number of UGEs during the follow-up, interval between UGE surveillance screens, the endoscopic and histopathological findings in the upper gastrointestinal tract, and *H. pylori* status.

Gastritis, atrophy, intestinal metaplasia, dysplasia, and *H. pylori* infection were evaluated according to the Sydney system classification [28]. Precancerous gastric lesions were defined as atrophic gastritis, intestinal metaplasia, or dysplasia. The *H. pylori* status was determined by histology or urea breath test.

### 2.3. Statistical Analysis

The prevalence of lesions was evaluated from the records of the first available UGE. For the calculation of incidence rates, all the patients included were considered to be at risk. The number of precancerous lesions or cancers that developed after the first UGE and the person-years of follow-up of each patient were used as the numerator and denominator, respectively, for estimating the incidence of precancerous lesions and cancers. The statistical associations between upper gastrointestinal lesion prevalence and MMR mutation, *H. pylori* infection, and age were analysed using the χ^2^ test. The incidence of gastrointestinal lesions was calculated for several different age classes: <30 years, 30–39 years, 40–49 years, 50–59 years, and >60 years. Statistical analyses were performed with Stata 9.0 statistical software (College Station, TX, USA). All tests were two-sided, and *p* values below 0.05 were considered to be statistically significant.

## 3. Results

### 3.1. Baseline Characteristics and Cumulative Endoscopic and Histopathological Findings

A total of 172 individuals undergoing this screening programme were evaluated. Their baseline characteristics are described in Table 2. The spectrum of endoscopic findings in our cohort is given in Table 3. One hundred and seventeen patients (68%) had undergone gastric biopsies during their initial endoscopic examination. Overall, 45 (26%) of the 172 patients were found to have precancerous lesions or cancer, 20 (44%) of whom had more than one lesion. Seventy neoplastic (precancerous or cancerous) upper gastrointestinal tract lesions (gastric, *n* = 59; duodenal, *n* = 11) were diagnosed. The dysplastic lesions and cancers were located in the stomach in 52% of cases and in the duodenum in the other 48% of cases. Most of the gastric lesions were distal (60%), including four of the five gastric cancers, which were located in the antrum. The cumulative frequencies of intestinal metaplasia, atrophic gastritis, dysplasia, and stomach cancer were 15%, 12%, 4%, and 3%, respectively. The cumulative frequencies of duodenal dysplasia and cancer were 4% and 3%, respectively.

### 3.2. Initial Upper Gastrointestinal Endoscopy Findings

The median age of patients at the first UGE was 44 years (range, 14–75). The initial UGE was performed for gastrointestinal symptoms in 24 patients (14%) detected during colonoscopy screening. Twenty-six of the 172 patients (15%) had precancerous lesions (*n* = 21, 12%) and/or cancer (*n* = 7, 4%) at their first endoscopy. Of the seven patients with gastric or duodenal cancer at initial UGE, five (63%) had digestive symptoms.

### 3.3. Endoscopic Follow-Up and Incidence of Upper Gastrointestinal Lesions

Follow-up UGE was performed in 109 of the patients (63%) included in this study. The median number of endoscopy procedures per patient was 3 (range, 2–10). The frequency of endoscopic surveillance was 1–2 years for 80 patients (73%), 3–4 years for 24 patients (22%), and 5 years or more for five patients (4.5%). Among the 63 patients who had undergone only one upper gastrointestinal endoscopy, 56 were included in the study less than 2 years after their first endoscopy, one patient had undergone total gastrectomy for gastric cancer, and only six patients had no gastrointestinal follow-up. Gastric biopsies were systematically included in the endoscopic examinations in 60 patients (55%) who underwent follow-up UGE.

After a mean follow-up of 5 years (range, 1–19), 19 patients were diagnosed with intestinal metaplasia (*n* = 15), gastric (*n* = 5) or duodenal (*n* = 5) dysplasia, atrophic gastritis (*n* = 13), gastric cancer (*n* = 1), or duodenal cancer (*n* = 1) after a median interval of 5 years since the initial UGE (range, 1–16). The incidence of gastric metaplasia, atrophic gastritis, gastric dysplasia, duodenal dysplasia, gastric cancer, and duodenal cancer was 26.0, 22.5, 8.7, 8.7, 1.7, and 1.7 per 1000 person-years, respectively. Of the 95 patients with normal findings at the initial UGE, none had cancer, and precancerous lesions were found in 19 (20%) of them during follow-up examinations. Of the ten patients with precancerous lesions at the initial examination, two (20%) developed cancer during follow-up. Seventy-six patients (70%) did not develop any precancerous lesion or cancer during follow-up.

### 3.4. Association of Upper Gastrointestinal Lesions with Baseline Characteristics

The incidences of precancerous and cancerous lesions by age group are presented in Figure 1. The median age of patients with precancerous lesions was 53 years (range, 23–71). The incidence of precancerous or cancerous upper gastrointestinal lesions was 6% among patients under 30 years of age, 14% among patients aged between 30 and 39 years, 27% among patients aged between 40 and 49 years, 53% among patients aged between 50 and 59 years, and 60% among patients aged between 60 and 69 years. The incidence of precancerous and cancerous upper gastrointestinal lesions was higher in patients over 40 years of age than in their younger counterparts (43% vs. 10%, *p* < 0.001).

The prevalence of precancerous and cancerous upper gastrointestinal lesions in *MHL1*, *MSH2*, and *MSH6* mutation carriers are presented in Figure 2. The only patient with a *PMS2* mutation had no lesion. Patients with an *MSH6* mutation had more intestinal metaplasia (9/26) than patients with an *MSH2* (9/82) or *MLH1* (8/63) mutation (*p* = 0.003). There was no statistically significant difference in the prevalence of other lesions; none of the patients with an *MSH6* mutation had upper gastrointestinal cancer.

No significant association was found between cancerous and precancerous lesions of the upper gastrointestinal tract and sex or family history of upper gastrointestinal cancer. Of the 26 patients with a family history of gastric cancer, six (23%) had gastric cancerous or precancerous lesions. Only one patient with gastric cancer had a family history of gastric cancer.

More specifically, of the 172 patients undergoing the screening programme, 10 (5.8%) had an upper gastrointestinal cancer. They were mainly males (66%) with a median age of 49 years (range, 14–66). All cancers were adenocarcinomas without independent cells (90%), except for one duodenal non-Hodgkin lymphoma. The characteristics of these adenocarcinomas are described in Table 4. There were five adenocarcinomas for each duodenal and gastric localization. One patient had a synchronous gastric and duodenal cancer; he carried an *MSH2* mutation and was treated with best supportive care. There were no significant differences in cancer characteristics between patients with *MLH1* and *MSH2* mutations even though there were numerically more duodenal adenocarcinomas in patients with an *MLH1* mutation (3 vs. 2). No cancer was detected in patients with *MSH6* or *PMS2* mutations. Seven patients had their cancer diagnosed at the initial UGE. Of these seven, five (63%) had digestive symptoms. All adenocarcinomas but one were surgically removed, including two after the initial mucosectomy. All adenocarcinomas but one had no distant metastasis and only three (30%) had lymph nodes spreading. *Helicobacter pylori* infection was confirmed by biopsy in only two patients (22%). These patients had both a localized gastric adenocarcinoma without lymph node spreading as well as an associated atrophic gastritis. Two patients had personal histories of endometrial cancer, including one carrying an *MSH2* mutation and with a personal history of colorectal and urinary tract cancers.

### 3.5. Helicobacter pylori Infection

Testing for *H. pylori* was performed in 145 patients (84%) by endoscopic biopsies (*n* = 143) or urea breath tests (*n* = 2). *H. pylori* infection was diagnosed in 41 patients (28%). The prevalence of precancerous and cancerous gastric lesions was higher in patients with *H. pylori* infection than in patients without *H. pylori* infection (51% vs. 13%, *p* < 0.001). *H. pylori* infection was not detected in any of the patients with precancerous or cancerous duodenal lesions. The prevalence of *H. pylori* did not differ significantly according to the type of MMR gene mutation. *H. pylori* infection was found in four of the five cases of gastric dysplasia diagnosed during surveillance, two of which had failed to respond to prior treatments intended to eradicate it.

## 4. Discussion

To the best of our knowledge, this is one of the largest studies to have assessed the prevalence and incidence of neoplastic lesions in individuals with Lynch syndrome enrolled in UGE screening programs. We found that the prevalence of gastroduodenal precancerous and cancerous lesions was high, which increased after 40 years and with *H. pylori* infection, but not with a family history of gastric or duodenal cancer. The results of our study suggest that individuals with Lynch syndrome should be considered for UGE screening, including gastric biopsies in order to check for preneoplastic lesions and *H. pylori* infection.

The link between Lynch syndrome and upper gastrointestinal cancer risk has been recognized for a long time. In spite of this longstanding association, there have been minimal data to date regarding the patient-specific factors that predict gastric cancer and duodenal cancer among Lynch syndrome carriers. Data from the multinational Prospective Lynch syndrome Database have suggested that older Lynch syndrome carriers with pathogenic *MLH1* or *MSH2* mutations are at particular risk for upper gastrointestinal cancers, with such individuals having a 7.1% and 7.7% cumulative risk of gastric cancer by age 75, respectively, with a 61% 5-year survival rate [7]. Other prior data have demonstrated that *MLH1* and *MSH2* carriers, especially males, are at higher risk for gastric cancer than female Lynch syndrome carriers and those with germline mutations in other Lynch syndrome genes [5,6,8]. The risk of duodenal carcinoma was reported to be the highest for the *MLH1* pathogenic variant carriers (6.5% for *MLH1* and 2% for *MSH2*), and no small bowel cancers were observed in patients with the constitutional *MSH6* or *PMS2* pathogenic variants [7]. Although nearly the majority of guidelines recommend considering an UGE with testing and treatment for *H. pylori*, the target Lynch syndrome population for screening, the age to begin UGE, and intervals for UGE follow-up are not consistent (Table 1). Because of a paucity of data, there is no concordance among the guidelines on which individuals with Lynch syndrome are most likely to benefit from screening. Conventional UGE is a minimally invasive procedure that can be performed during the same general anaesthesia that the screening colonoscopy regularly accomplished in Lynch syndrome patients. Despite this fact, there are only few studies that have evaluated the effectiveness of screening and surveillance for upper gastrointestinal cancer in these patients.

Lynch syndrome-related colon cancers appear to follow an accelerated colonic adenoma–carcinoma sequence [29]. Lynch syndrome-related upper gastrointestinal cancers could result from a similar sequence, with accelerated progression of dysplasia to cancer. The majority of gastric cancers in Lynch syndrome patients appear to be histologically classified as the intestinal type and, consequently, potentially amenable to endoscopic surveillance [5,22]. Lynch syndrome patients develop rare gastric adenomatous polyps compared to patients with familial adenomatous polyposis or MYH associated polyposis. In these latter hereditary syndromes, fundic gland polyp is the most common gastric abnormality. It can develop dysplasia and rarely into invasive adenocarcinoma. In Lynch syndrome, no clear specific polypoid precancerous pattern has been described even though it has been suggested that pyloric gland adenomas may be precursors of gastric cancer [30]. In a young cohort of Lynch syndrome patients, chronic immune gastritis was observed in over 70% of cases with gastric cancer and none of the patients without gastric cancer [31]. *H. pylori* infection has been reported in about 20% of gastric cancers in Lynch syndrome, which is in adequation with our study [22,32]. These findings suggest that an evaluation for underlying chronic gastritis as well as *H. pylori* infection should be considered in surveillance protocols for younger individuals with Lynch syndrome. The prevalence of atrophic gastritis and intestinal metaplasia in our study was in accordance with a previous study by Renkonen et al., reporting a prevalence of 14% [23]. However, we found 10 cases of cancer and 13 cases of low- or high-grade dysplasia, 80% and 54% of which, respectively, were detected at the first endoscopic exam, versus one case of duodenal cancer and no case of dysplastic lesion in this previous study. This difference may be explained by the fact that our study included patients who already had digestive symptoms, as was the case for six out of the nine patients with cancer. Moreover, the study by Renkonen et al. included 73 patients who had undergone endoscopic examination only once and may therefore have underestimated the frequency of neoplastic lesions. Another small, retrospective, single-institution study of 66 LS carriers (21 of whom underwent EGD surveillance) found that 19% of the screened individuals had potential precursor findings on UGE (*H. pylori*, intestinal metaplasia) [24]. Our results are contrary to a more recent single-institution cohort of 217 LS carriers whom underwent UGE surveillance, where the prevalence of gastric intestinal metaplasia was low (8.3%) [26]. The prevalence of *H. pylori* infection was 28% in our cohort, which is consistent with the usual prevalence in Europe (around 30%) [33]. In our study, the prevalence of gastric precancerous lesions and cancer was significantly higher in patients with *H. pylori* infection (51% vs. 13%, *p* < 0.001). Our results indicate that surveillance programs in Lynch syndrome carriers should include investigation of *H. pylori* infection. As in previous studies, we found no significant difference between *MLH1* and *MSH2* mutation carriers [5,34]. In contrast, the prevalence of intestinal metaplasia was significantly higher in *MSH6* mutation carriers than in MLH1 or *MSH2* carriers. This difference was not related to the corresponding variation in the prevalence of *H. pylori* infection, which was similar in carriers of the different mutations. The reasons for such a difference (which need to be confirmed by further studies) are unclear. Although some experts propose an endoscopy if more than one family member have gastric cancer, there is no clear evidence that a family history indicates a predisposition in Western countries [2,5]. In a German cohort study of 988 individuals with Lynch syndrome, only one-quarter of patients with gastric cancer had a family history of gastric cancer [34]. A recent study reported that the reported number of first-degree relatives with gastric cancer were independently associated with a reported personal history of gastric cancer in Lynch syndrome patients [35]. In our study, there was no difference in the prevalence of gastroduodenal precancerous lesions or cancers related to family history.

The effectiveness of surveillance using UGE in patients with Lynch syndrome remains controversial, and the recommendation varies among regions [36]. One of the reasons for the controversy is the geographical variation of gastric cancer incidence. As in sporadic cases, the incidence of gastric cancer in Lynch syndrome is higher in Asian countries (up to 40%) compared with in Western countries (ranging from 1 to 10%) [4,5,7,36,37,38,39]. Based on these reports, environmental factors and particularly *H. pylori* infection act as risk factors for gastric cancer. In 2015, the worldwide prevalence for *H. pylori* infection was 4.4 billion individuals, with a wide disparity according to geographic regions [40]. In sporadic cases, *H. pylori* is the main cause for chronic gastritis and the main etiological agent for peptic ulcer disease and gastric cancer [41,42]. These clinical outcomes result in a complex multifactorial interplay between bacterial virulence, host immune response, and microbiota modifications [43]. Approximately 89% of all gastric cancers can be attributable to *H. pylori* infection [44]. For these reasons, *H. pylori* has been identified as a Group 1 carcinogen by the International Agency for Research on Cancer and is considered a necessary but insufficient cause of gastric adenocarcinoma. According to Soer et al., it seems that the prevalence of *H. pylori* infection rates in Lynch syndrome mutations carriers does not differ from the general population [32]. Based on these reports, *H. pylori* infection significantly affects gastric cancer risks in patients with Lynch syndrome. Its eradication is a public health issue of main interest needing more efficient screening and therapeutic strategies. A major hurdle to eradication is increasing antibiotic resistance [45]. To that concern, clear guidelines for the evaluation and treatment of *H. pylori* infection must be regularly put up to date [45]. In our study, the only two patients diagnosed with gastric adenocarcinoma and *H. pylori* infection were also the only ones presenting atrophic gastritis. Patients with Lynch syndrome, particularly those with atrophic gastritis, are at high risk of gastric dysplasia and cancer, and the occurrence of synchronous and/or metachronous multiple lesions is common [46]. Some reports suggested that endoscopic surveillance of Lynch syndrome patients for gastric cancer should be provided based on the estimated risk in each patient [7,13]. In our study, most of the patients underwent upper gastrointestinal endoscopy concomitantly with colonoscopy at 1- to 2-year intervals. However, there is no consensus about the appropriate screening for upper gastrointestinal cancer in Lynch syndrome (Table 1). Guidelines are all based on imprecise or small-sized epidemiological studies, due to the lack of UGE surveillance studies [24,25,26]. Nevertheless, Capelle et al. have reported that, in a large Dutch cohort of individuals with Lynch syndrome, gastric cancer cases were mostly diagnosed after 45 years and that a family history of gastric cancer was a poor prognosticator of an increase in individual risk [5]. They therefore suggested starting gastric surveillance at 45 years of age, regardless of family history. In our study, however, some precancerous lesions were detected during the third decade of life. The clinical benefit of screening Lynch syndrome patients with gastroscopy probably exists. The decrease in annual incidence and relative proportion for gastric cancer in the general population in later Lynch generations raises the issue of the value of UGE screening in Lynch families [47,48]. A recent large study supports the recommendations of regular UGE surveillance for Lynch syndrome patients. In a prospective, multicentre cohort study of 1128 Lynch syndrome patients, gastric cancers in patients undergoing regular UGE every 1–3 years were significantly diagnosed, and this more often in an early stage of disease than gastric cancers detected through symptoms (83% vs. 25%) [27]. The limited data on the efficacy of UGE screening is a real-world clinical dilemma for Lynch syndrome-associated upper gastrointestinal cancers. We therefore suggest that an initial UGE with systematic gastric biopsies should be performed when Lynch syndrome is diagnosed in order to check for gastric or duodenal precancerous lesions and *H. pylori* infection. Our results do not support a systematic policy of examinations at 1- to 2-year intervals for upper gastrointestinal endoscopy for all individuals with Lynch syndrome. Firstly, the risk of gastric cancer seems to be low if no precancerous lesion or *H. pylori* infection is detected by the first endoscopy. Secondly, 70% of patients with endoscopic surveillance did not go on to develop precancerous lesions or cancer. Endoscopic surveillance might therefore be offered to individuals with precancerous lesions detected at their first endoscopy. Even if the first endoscopy is normal, it seems reasonable to offer endoscopy after the age of 40 years as the risk of cancer increases after this age.

Screening for small-bowel cancer is currently not recommended. Studies of small bowel screening in Lynch syndrome patients are lacking. A recent study suggests regular duodenal screening during UGE, especially in *MSH2* patients [25]. Since half the cases of small-bowel cancer in their cohort occurred in the duodenum, Schulmann et al. suggested adding UGE or push enteroscopy to screening colonoscopy in individuals with Lynch syndrome after the age of 30 years [11]. Guidelines do not recommend surveillance for small bowel cancer by videocapsule or enteroscopy but do suggest inspecting the distal duodenum during UEG if this is performed.

Our study has some limitations, including that our relatively small sample size limits the capacity to detect risk factors among patient with developed upper gastrointestinal cancer and to identify an optimal UGE screening interval. There also might be a bias in our study population. Universal tumour testing for Lynch syndrome revealed that *MSH6* and *PMS2* carriers are more frequent than previously recognized [49]. Most of the registered Lynch syndrome patients in our study were *MLH1* and *MSH2* carriers, which are potentially at high risk of Lynch syndrome-associated cancers [8]. Thus, we cannot exclude the possibility of overestimation of upper gastrointestinal cancer risk compared with the genuine Lynch syndrome population. Information on *H. pylori* status and histological assessment of gastric mucosa metaplasia was not available for all patients. However, both endoscopic findings of gastric atrophy and histological assessment of intestinal metaplasia were reported to be an accurate alternative for atrophic gastritis with excellent interobserver agreement [50,51]. As atrophic gastritis has been reported to be strongly associated with *H. pylori*, we consider that the endoscopic and histological findings can be a good substitute for *H. pylori* status in clinical settings [52]. Our findings underline the crucial need for future larger collaborative studies involving healthcare networks specialized in Lynch Syndrome management to scrutinise this issue.

## 5. Conclusions

In conclusion, our study may help to work out an appropriate screening program for gastric and duodenal cancer in individuals with Lynch syndrome. We show that the incidence of gastroduodenal precancerous lesions and cancers increased significantly after 40 years and was associated in more than 50% of patients with *H. pylori* infection. Nevertheless, the incidence of dysplastic lesions and cancer was low during endoscopic follow-up. Although we emphasize that our data are insufficient to justify a specific approach to UGE screening, we suggest a first UGE in all patients with Lynch syndrome with systematic gastric biopsies to check for precancerous lesions and screening for *H. pylori* infection, regardless of the type of MMR gene mutation or any family history of gastroduodenal cancer. During follow-up, UGE could be offered to all Lynch syndrome patients with a non-*PMS2* mutation, a family history of upper gastrointestinal cancer, or upper gastrointestinal precursor lesions such as gastric atrophy, autoimmune gastritis, intestinal neoplasia, or duodenal adenoma, while recognizing the limitations of the current evidence. Future studies should assess the effectiveness and cost-effectiveness of periodic (e.g., biennial) UGE surveillance for patients with precancerous lesions at the first endoscopy and for individuals over 40 years or with others risk factors, and whether longer surveillance intervals could be safe in other individuals with Lynch syndrome.

## Figures and Tables

**Figure 1 cancers-13-01657-f001:**
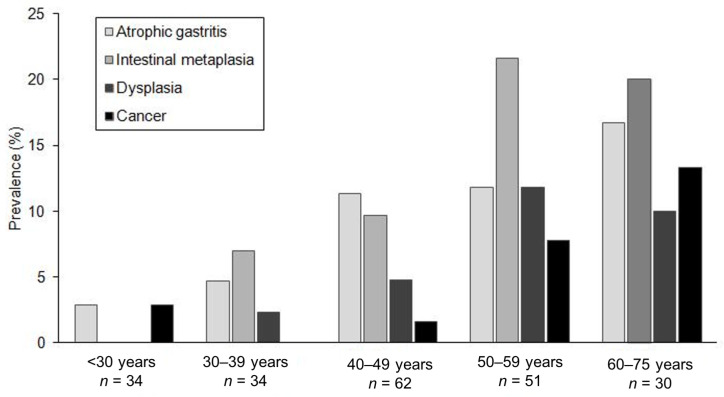
Incidence of gastric precancerous lesions and cancer by age group.

**Figure 2 cancers-13-01657-f002:**
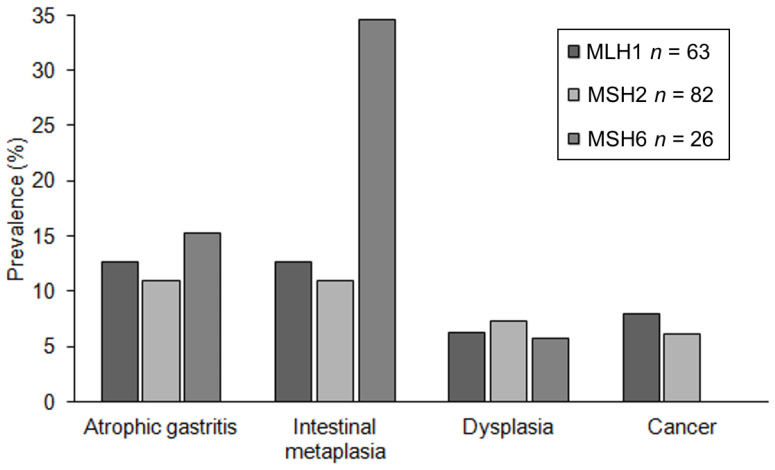
Prevalence of precancerous lesions and cancer in carriers of the MHL1, MSH2, and MSH6 mutations.

**Table 1 cancers-13-01657-t001:** Current upper gastrointestinal screening guidelines from various medical societies in Lynch syndrome.

Medical Society	Guidelines
United States Multi-Society Task Force [13]	- Consider baseline upper gastrointestinal endoscopy with biopsy for *H pylori* for all Lynch syndrome carriers at age 30–35 - Consider ongoing surveillance every 2–3 years based on individual patient risk factors
American College of Gastroenterology [15]	- Consider baseline upper gastrointestinal endoscopy with biopsy for *H. pylori* for all Lynch syndrome carriers at age 30–35 - Consider ongoing surveillance every 3–5 years for Lynch syndrome carriers with a family history of gastric or duodenal cancer
American Society of Clinical Oncology [16]	- Consider upper gastrointestinal endoscopy surveillance every 1–3 years in high-risk subsets of Lynch syndrome carriers - Test all Lynch syndrome carriers for *H. pylori*
European Society of Digestive Oncology and European Society for Medical Oncology [17]	- Upper gastrointestinal endoscopy should be performed regularly every 1–2 years in mutation carriers starting no later than the age of 30 years, regardless of the family history - Test all Lynch syndrome carriers for *H. pylori*
British Society of Gastroenterology and Association of Coloproctology of Great Britain and Ireland and United Kingdom Cancer Genetics Group [18]	- We recommend that gastric or small bowel surveillance in Lynch syndrome patients is only performed in the context of a clinical trial. - We recommend screening for *H. pylori* in patients with Lynch syndrome and subsequent eradication therapy if indicated.
European Hereditary Tumour Group and European Society of Coloproctology [14]	- Consensus was not achieved for the statement “Surveillance for other cancers (than colorectal, endometrial and ovarian) should not be offered”.

**Table 2 cancers-13-01657-t002:** Baseline patient characteristics.

Characteristic	Patients *n* (%) (*n* = 172)
Sex, male/female	66/106 (38/62)
MMR * genes mutation	
*MLH1*	63 (37)
*MSH2*	82 (48)
*MSH6*	26 (15)
*PMS2*	1 (<1)
Family history of gastric cancer	26 (15)
Family history of duodenal cancer	8 (5)
Personal history of cancer	84 (49)
Median age at diagnosis of first cancer (range)	41 (14–61)
Personal history of cancers	
Colorectal cancer	73 (42)
Endometrial cancer	15 (9)
Urinary tract cancer	5 (3)
Ovarian cancer	5 (3)

* MMR: Mismatch repair.

**Table 3 cancers-13-01657-t003:** Cumulative endoscopic findings in our study population.

Endoscopic Findings		At First UGE (*n*)	At Later UGE (*n*)	Total Patients: *n* (% of *n* = 172)
Cancers	Gastric adenocarcinoma	4	1	5 (3)
	Duodenal adenocarcinoma	4	1	5 (3)
	Duodenal non-Hodgkin lymphoma	0	1	1 (<1)
Gastric polyps	Adenomatous polyps	1	3	4 (2)
	Low-grade dysplasia	1	2	3 (2)
	High-grade dysplasia	0	1	1 (<1)
	Fundic gland polyps	9	4	13 (8)
Duodenal polyps	Adenomatous polyps	1	5	6 (2)
	Low-grade dysplasia	1	4	5 (3)
	High-grade dysplasia	0	1	1 (<1)
	Hyperplasic polyps	1	0	1 (<1)
	Inflammatory polyps	1	0	1 (<1)
Other gastro-duodenal lesions	Atrophic gastritis	8	13	21 (12)
	Intestinal metaplasia	11	15	26 (15)
	Low-grade dysplasia	1	1	2 (1)
	High-grade dysplasia	0	1	1 (<1)

**Table 4 cancers-13-01657-t004:** The characteristics of patients with adenocarcinoma in our study.

Characteristics	Patients with *MLH1* Mutation (*n* = 5)	Patients with *MSH2* Mutation (*n* = 4)	Total Patients with Adenocarcinomas (*n* = 9)
Median age: years (range)	40 (14–54)	51 (42–61)	49 (14–61)
Gender: *n* (%)			
Male	3 (60)	3 (75)	6 (67)
Female	2 (40)	1 (25)	3 (33)
Personal history of cancers:			
Colorectal cancer	0 (0)	1 (25)	1 (11)
Endometrial cancer	1 (20)	1 (25)	2 (22)
Urinary tract cancer	0 (0)	1 (25)	1 (11)
Diagnosed cancers at initial UGE: *n* (%)	4 (80)	3 (75)	7 (78)
Digestive symptoms at initial UGE: *n* (%)	3 (60)	2 (50)	5 (63)
Tumour localization: n (%)			
Gastric	2 (40)	3 (60)	5 (50)
Duodenal	3 (60)	2 (40)	5 (50)
Differentiation grade: *n* (%)			
Well	1	1	2 (20)
Moderate	2	2	4 (40)
Poor	0	0	0 (0)
Missing data	2	1	3 (30)
TNM UICC 2016 staging at diagnosis: *n* (%)			
I	1	2	3
II or III	2	1	3
IV	1	0	1 (10)
Lymph node spreading: *n* (%)	2	1	3 (30)
Curative surgery: *n* (%)	5 (100)	3 (75)	9 (90)
*Helicobacter pylori* infection: *n* (%)	1 (20)	1 (25)	2 (22)
Associated atrophic gastritis	1 (20)	1 (25)	2 (22)

## Data Availability

The data presented in this study are available on request from the corresponding author.

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
