# Peer review of "Upper Gastrointestinal Lesions during Endoscopy Surveillance in Patients with Lynch Syndrome: A Multicentre Cohort Study"

_cancers, 2021, doi:10.3390/cancers13071657_

Round 1

Reviewer 1 Report

Chautard et al presented the results of UGE in a cohort of Lynch syndrome patients.

Although the number of patients is low, compared to other recent studies, the study is interesting.

Comments:

Major points:

The authors should include, in addition to the specific clinical data, pathological data of the cancers detected in both the stomach and duodenum: histological type, grade, stage, MMR protein analysis, etc. These data are of great interest for the readers. The authors should include a specific section for these results in the text and an additional table.

Minor points:

Two tables 1 are presented.

In the first table 1 is difficult to read which criteria corresponded to each guideline.

The Table 2 should be modified to introduce a column corresponding to lesions detected at baseline and another one to lesions detected in the follow-up

Table 2 includes duodenal adenomatous polyps not referred in the text. Do they referred to duodenal dysplasia in the text? Please, clarify.

Author Response

Dear reviewer, 
We would like to thank your for your comments to improve the scientific quality of our manuscript. Our answers are given below. The corrections in the manuscript will be highlighted in yellow.

Comments :
Major points:
The authors should include, in addition to the specific clinical data, pathological data of the cancers detected in both the stomach and duodenum: histological type, grade, stage, MMR protein analysis, etc. These data are of great interest for the readers. The authors should include a specific section for these results in the text and an additional table.
Minor points:
Two tables 1 are presented.
In the first table 1 is difficult to read which criteria corresponded to each guideline.
The Table 2 should be modified to introduce a column corresponding to lesions detected at baseline and another one to lesions detected in the follow-up
Table 2 includes duodenal adenomatous polyps not referred in the text. Do they referred to duodenal dysplasia in the text? Please, clarify.

Answers : 
- We have added a specific table (table 3) and paragraph (included in section 3.5) concerning pathological data of the detected cancers;
- We have modified tables 1 and 2 for respectively guidelines to be more readable and endoscopic findings to be easily interpreted according to their type and time of UGE;

We sincerely hope that these modifications will satisfy your advised review.

Best regards,

Thierry Lecomte and Romain Chautard 

Reviewer 2 Report

The article "Upper gastrointestinal lesion during endoscopy surveillance in patients with Lynch syndrome: a multicentre cohort study", by Romain Chautard et al. provides new information of upper  gastrointestinal lesions and cancers in Lynch syndrome patients. The article it self is very well written and it is easy to follow and read. However a few things needs to correct and/or clarify.

In the Simple Summary: ...cohort of 172 Lynch syndrome CARRIERS??  I would use either Lynch syndrome patients or Lynch syndrome mutation carriers.

In the Introduction: In the end of first section ("...risk of endometrial cancer"), there is no reference at all. Even though I am not sure whether next references [2, 4-6] also are meant for this section as well, but it would be good to have a reference/s here as well.

In the table 1, this would benefit at least row lines to separate different Guidelines to each other.

There are two table 1, First table and the second table are both Table 1!

Finally, even though this article findings are based on Lynch syndrome, I think that comparison with sporadic/reference cases would give a deeper perspective for readers. I do not expect comparison to made all Table 2 (3) Endoscopic findings, but at least cancers and the most common polyps.

Also, I think that it would be good to mention also how common the Helicobacter Pylori infection is in general and its incidence and evaluated connection in gastric cancer in sporadic population. 

Author Response

Dear reviewer, 

We would like to thank your for your comments to improve the scientific quality of our manuscript. Our answers are given below. The corrections in the manuscript will be highlighted in yellow.

Comments: 

In the Simple Summary: ...cohort of 172 Lynch syndrome CARRIERS??  I would use either Lynch syndrome patients or Lynch syndrome mutation carriers.

In the Introduction: In the end of first section ("...risk of endometrial cancer"), there is no reference at all. Even though I am not sure whether next references [2, 4-6] also are meant for this section as well, but it would be good to have a reference/s here as well.

In the table 1, this would benefit at least row lines to separate different Guidelines to each other.

There are two table 1, First table and the second table are both Table 1!

Finally, even though this article findings are based on Lynch syndrome, I think that comparison with sporadic/reference cases would give a deeper perspective for readers. I do not expect comparison to made all Table 2 (3) Endoscopic findings, but at least cancers and the most common polyps.

Also, I think that it would be good to mention also how common the Helicobacter Pylori infection is in general and its incidence and evaluated connection in gastric cancer in sporadic population. 

Answers:

- We rephrased the sentence In the Simple Summary.

- We have added a reference for the risk of endometrial cancer in the end of the first section of the introduction;

- We have modified table 1 for guidelines to be more readable;

- We have corrected the numbering of the tables;

- We have added comments with references concerning specific pre-cancerous polyps in Lynch syndrome patients in the third paragraph of the discussion;

- We have added comments with references in the fourth paragraph of the discussion, concerning Helicobacter pylori infection in general and its correlation with gastric cancer in sporadic and Lynch syndrome patients;

We sincerely hope that these modifications will satisfy your advised review.

Best regards,

Thierry Lecomte and Romain Chautard

Round 2

Reviewer 1 Report

No comments.

Reviewer 2 Report

Dear authors of "Upper gastrointestinal lesions during endoscopy surveillance in patients with Lynch syndrome: a multicentre cohort study", I am pleased to noticed all the corrections you have made. I have no other correction suggestions to this current form of your manuscript.